# In Silico Structural Prediction for the Generation of Novel Performant Midi-Dystrophins Based on Intein-Mediated Dual AAV Approach

**DOI:** 10.3390/ijms251910444

**Published:** 2024-09-27

**Authors:** Laura Palmieri, Maxime Ferrand, Ai Vu Hong, Isabelle Richard, Sonia Albini

**Affiliations:** 1Genethon, 91000 Evry, France; lpalmieri@genethon.fr (L.P.); ferrand@genethon.fr (M.F.); avuhong@genethon.fr (A.V.H.); richard@genethon.fr (I.R.); 2INTEGRARE Research Unit UMR_S951 (INSERM, Université Paris-Saclay, Univ Evry), 91000 Evry, France; 3Atamyo Therapeutics, 1, Bis Rue de l’Internationale, 91000 Evry, France

**Keywords:** DMD, intein, midi-dystrophin, dual-AAV, AlphaFold, structural prediction

## Abstract

Duchenne Muscular Dystrophy (DMD) is a pediatric disorder characterized by progressive muscle degeneration and premature death, and has no current cure. The current, most promising therapeutic avenue is based on gene replacement mediated by adeno-associated viruses (AAVs) using a shortened, but still functional, version of dystrophin, known as micro-dystrophin (µDys), to fit AAV capacity. The limited improvements observed in clinical trials suggest a sub-optimal performance of µDys in the human context that could be due to the lack of key domains in the protein. Therefore, expressing larger dystrophin proteins may be necessary for a more complete correction of the disease phenotype. In this study, we developed three novel midi-dystrophin constructs using a dual-AAV approach, leveraging split-intein-based protein trans-splicing. The midi-dystrophins include additional domains compared to µDys, such as the central cytoskeleton-binding domain, nNOS and Par1b interacting domains, and a complete C-terminal region. Given the limited capacity of each AAV vector, we strategically partially reduced hinge regions while ensuring that the structural stability of the protein remains intact. We predicted the interactions between the two halves of the split midi-Dys proteins thanks to the deep learning algorithm AphaFold3. We observed strong associations between the N- and C-termini in midi-Dys 1 and 2, while a weaker interaction in midi-Dys 3 was revealed. Our subsequent experiments confirmed the efficient protein trans-splicing both in vitro and in vivo in DBA2/mdx mice of the midi-Dys 1 and 2 and not in midi-Dys 3 as expected from the structural prediction. Additionally, we demonstrated that midi-Dys 1 and 2 exhibit significant therapeutic efficacy in DBA2/mdx mice, highlighting their potential as therapeutic agents for DMD. Overall, these findings highlight the potential of deep learning-based structural modeling for the generation of intein-based dystrophin versions and pose the basis for further investigation of these new midi-dystrophins versions for clinical studies.

## 1. Introduction

Duchenne Muscular Dystrophy (DMD) is a recessive X-linked disease that affects one in 3500 to 5000 boys [1]. Dystrophin plays a major role in the mechanical support of muscle fibers and its absence results in muscle degeneration, characterized by a progressive muscle wasting leading to a loss of walking around the teenage years. The respiratory and heart muscles also gradually weaken, causing premature death [2,3].

The dystrophin protein can be divided into four structural regions: the actin-binding domain (ABD) at the N-terminus, the central rod domain, the cysteine-rich domain (CR), and the C-terminal domain (C-term). The central rod domain is formed by 24 spectrin-like regions (R), intercalated by 4 intrinsically disordered regions called hinges (H). In skeletal muscle fiber, dystrophin interacts with several cytoskeletal proteins, particularly actin, microtubules, and intermediate filaments, as well as with sarcolemma lipids [4,5,6,7,8]. Additionally, thanks to the CR, dystrophin interacts with the dystroglycan complex (DGC), a transmembrane group of proteins which in turns bind to the laminin of the extracellular matrix (ECM) [2,9]. These interactions create a structural bridge between the cytoskeleton and the ECM, providing support to the myofibers and protecting the skeletal muscle cell from contraction-induced damage. Additionally, dystrophin interacts with several signaling proteins, such as neuronal nitric oxide (nNOS) [10] and Par1b, a serine/threonine kinase that controls cell polarity and cell–cell interaction [11]. Moreover, the C-terminal region of dystrophin binds to syntrophin and dystrobrevin, thus ensuring the formation and maintenance of neuromuscular junctions, as well as stabilizing the binding with nNOS [12]. All these interactions contribute to the maintenance of the homeostasis of the skeletal muscle fiber.

While DMD remains a fatal and incurable disease, a variety of promising gene therapies mediated by adeno-associated virus (AAV) have been recently translated into clinical trials. These approaches restore the sarcolemma stability of dystrophic myofibers through the expression of shorter versions of dystrophin (µDys), which contain the domains necessary to connect the actin filaments and the DGC [13,14,15]. The recent accelerated approval of Sarepta’s µDys by the US Food and Drug Administration (FDA) has brought hope to patients aged 4 and more [16]. Nevertheless, due to the lack of critical domains necessary for full dystrophin function, µDys gene therapies are de facto limited in their ability to fully restore the homeostasis of skeletal and cardiac tissues [13,14,17,18]. In view of this limitation, we decided to generate larger versions of dystrophin, including specific additional domains.

Several attempts have been made to deliver longer versions of dystrophin by harnessing dual- or triple-AAV approaches [12,19,20,21,22,23,24,25,26]. Various reconstitution mechanisms were employed, including cDNA trans-splicing, homologous recombination (HR), and mRNA and protein trans-splicing. The transfer of midi-dystrophins or full-length dystrophin using cDNA trans-splicing [20,27,28] and overlapping HR [19,21,22] successfully demonstrated therapeutic efficacy in dystrophin-null mice. However, these approaches showed around 25 to 50% efficacy of reconstitution [19,20,21,29], thus requiring high doses of AAVs with the potential to be associated with immune response and toxicities [30]. Another method for creating larger proteins from smaller transcription units involves utilizing split-inteins to fuse two proteins. Split-inteins are small polypeptides that undergo a distinct post-translational process called protein trans-splicing (PTS). During PTS, the N- and C-terminal residues (exteins) flanking the inteins are naturally ligated, resulting in the formation of one complete protein while removing the reconstituted inteins [31]. Intein-mediated PTS has been widely explored in the context of retinal and liver diseases [32,33,34] and recently in the context of DMD [23,26]. Specifically, the delivery of intein-based dual-AAVs in dystrophin-null mice demonstrated nearly 100% PTS efficiency for midi-dystrophins [23] and 60% efficiency for reconstituting full-length dystrophin [26]. These approaches led to a significant improvement of the dystrophic phenotype and physiological correction in young and old mdx mice.

Here, we generated three novel midi-dystrophins using a split-inteins dual-AAV approach based on Artificial Intelligence (AI) structural prediction to monitor the impact of the changes made during the design of the protein structure. Because of the limited capacity of AAV, we strategically decided to reduce the size of the hinges to make room for additional functional domains. We then assessed the correct 3D conformation using AI tool upon these changes. Additionally, we digitally simulated the interactions between the two halves of the split midi-Dys proteins and observed strong interactions between the N- and C-termini in midi-Dys 1 and 2, with a weaker association in midi-Dys 3. We demonstrated high protein trans-splicing (PTS) efficiency using an in vitro GFP reporter system and in vivo studies in DBA2/mdx mice, with approximately 90% PTS efficiency for midi-Dys 1 and midi-Dys 2, while midi-Dys 3 exhibited lower reconstitution efficiency. Consistent with this, we demonstrated a high therapeutic efficacy of midi-Dys 1 and 2 in vivo in DBA2/mdx mice.

## 2. Results

### 2.1. Design of Three Novel Midi-Dystrophins Proteins Based on Intein-Mediated Reconstitution Approach

The rationale behind the design of the new midi-dystrophins was based on evidence in the literature about the crucial domains for dystrophin function [4,5,6,8,9,11,12,35]. Because of the importance of the dystrophin signaling role, we decided to include the binding domains for Par1b (R8-9), nNOS (R16-17) and syntrophin/dystrobrevin (C-term) in all the three forms designed. However, we varied the inclusion of domains involved in dystrophin’s interaction with structural elements of the myofibers. Specifically, binding to membrane lipids (R1-3, R10-12) was fully included in midi-Dys 1 and 3, but only partially in midi-Dys 2. Interactions with microtubules (R4-15, R20-24) were included in midi-Dys 2 and 3, and partially in midi-Dys 1, while the internal actin-binding site (R11-17) was partially included in midi-Dys 1 and 2, and not in midi-Dys 3. The ABD1 (N-term) and the binding with β-Dystroglycan are common in all our constructs (Figure 1A–C). To make more space for key structural domains, we strategically sacrificed part of the hinges which link the several spectrin-like repeats in the central domain, and which are intrinsically disordered regions of dystrophin. In all our constructs, we shortened the hinges by 36 to 56%, where the most important reduction was applied to hinge 1, where we eliminated the first 54 amino acids (Appendix A). To determine if our modifications to the primary amino acid sequence caused abnormal protein conformation, we predicted the 3D structure of the designed midi-dystrophins. To do that, we employed AlphaFold3, an AI-based deep learning algorithm that can predict protein structures with atomic accuracy, even in cases in which no similar structure is known [36]. Since the prediction with AlphaFold can only yield 2000 amino acids, we trimmed the three midi-dystrophins in three parts: the N-terminal (which includes the ABD1, the H1, and the R1), the central domain (which includes the R1 to the R24) and the C-terminal domain (which includes the R24, the H4, the Cystein rich domain, and the C-terminal). It is worth noting that the first and the last regions share commonalities with each other in the three sequences. Firstly, these three parts were compared to the prediction obtained with the original untruncated hinge sequences. As depicted in Appendix A, no change in the structural conformation of the N-terminal and C-terminal regions was observed after the shortening of the hinges. Moreover, no significant alterations in the alpha-helix structure of the spectrin-like repeats and in the 3D conformation were detected with the inclusion of the shortened hinges (Figure 1D–F). The predicted template modeling (pTM) score, which measures the accuracy of the entire protein structure [37], is not significantly different between the structural prediction of the original and shortened hinges (Figure 1G). This result indicates that the shortening of these regions does not affecting the 3D conformation of the proteins.

Once we determined the aminoacidic sequence of the structures, we evaluated the possible site where inteins could be inserted. Based on a previously described intein library screening [38], we selected GP41-1 as the most performant intein in terms of speed of reaction and efficiency of protein trans-splicing ability. Additionally, GP41-1 is one of the shortest inteins known so far, thereby allowing more space to include functional domains in our constructs. Intein activity is context-dependent, particularly to the aminoacidic environment that surrounds their ligation junction (N- and C-exteins). For GP41-1, its native extein junction sequence is SGY/SSS, where the most crucial residue for the PTS reaction is the first amino acid in the C-extein, which is a Serine [39,40] (Figure 1H). Additionally, it was shown that the deletion of the native extein residues does not impact the cleavage reaction yields of GP41-1 [39], which makes it possible to insert the intein in the protein sequences without including additional residues on the targeted proteins. Based on this evidence, we split the midi-dystrophin protein sequences, maintaining a Serine in the first amino acid position the C-extein and respecting the limited capacity of 4.7 kilobases of the rAAV, where possible. Specifically, the GP41-1 was inserted in R11 and R17 in midi-Dys 1 and midi-Dys 2, respectively, while for midi-Dys 3 it was inserted into hinge 3.

### 2.2. In Silico Modeling of Intein Adjacent Sites Predicts Efficacy of Protein Reconstitution

The efficacy of split-intein protein trans-splicing is dependent on the association and folding of the split halves, influencing the ligation kinetics [41]. We therefore hypothesize that the residue interactions of the domains around intein subunits may influence the protein trans-splicing efficiency. To investigate these interactions, we performed a structure prediction of a GFP split-intein reporter by using AlphaFold3 [36]. We designed two plasmids, each encoding either the N- and the C-terminal half of the reporter GFP fused to the N- and C-terminal halves of the GP41-1 intein under the control of CK8 promoter. Additionally, an HA tag was added to the C-terminal end of the GP41-1 N-terminal half (Figure 2A). We initially aligned the GP41-1 structure predicted using AlphaFold3 for GFP with the structure obtained from the intein subunits alone (without any surrounding sequences). The two structures displayed identical conformations, with a Pymol root-mean-square deviation (RMSD) of 0.226. Furthermore, the intein-associated GFP structure demonstrated proper folding, with an RMSD of 0.154 compared to the intact, unsplit GFP. To evaluate the residue interaction within the intein subunits, we performed a structural prediction focusing on the GP41-1 intein. The GFP-associated intein showed high confidence of structure, with a predicted local distance difference test (pLDDT)_Intein-GFP_ of 89.09 (close to the 90 cut-off of high accuracy), and high confident residue relative positions, with a Predicted Aligned Error (PAE), a metric measuring interdomain structural error, of 2.06 Å (Figure 2B–D). The two GFP halves (colored in light cyan and light pink) in the predicted structure interact with each other (Figure 2C). Indeed, the PAE showed a highly confident interaction of the GFP N- and C-termini subunits, with 77.92% of the pairwise inter-domain PAE less than 10Å (%PAE_<10Å_) (Figure 2D). The prediction of the GFP split-intein 3D structure indicated the correct incorporation of the intein, suggesting the possibility of high excision activity.

To actually evaluate intein protein reconstitution efficacy in vitro, we transfected HEK293T cells with either single or dual plasmids GFP-GP41-1 split-intein, with, as a control, a plasmid encoding for the complete protein sequence of GFP under the control of the CK8 promoter (Figure 2E). Seventy-two hours after transfection, GFP fluorescence was observed in the dual plasmid condition, albeit with a reduced intensity compared to the GFP plasmid, while no fluorescence was observed in the single-split controls. Via fluorescence-activated cell sorting (FACS), we found that the GFP fluorescence intensity was 7.7 × 10^5^ in the dual GFP condition, while we obtained a GFP fluorescence of 1.1 × 10^6^ in the single plasmid GFP condition (Figure 2F). Based on these results, we confirm the efficacy of the reconstitution of the intein GP41-1 in the GFP construct.

### 2.3. Structural Predictions of GP41-1 Split Midi-Dys Revealed Efficiency of Protein Trans-Splicing

To determine if the residues interaction in the intein adjacent sites influences the protein trans-splicing efficiency of our midi-dystrophins, we performed a structure prediction of the complexes containing 200 amino-acids (aa) of exteins around the split-intein protein by using AlphaFold3 [36]. Within all three midi-Dys complexes, the two split-intein subunits (colored in cyan and pink) showed almost identical structures (RMSD_Intein-MD1/Intein_ = 0.189, RMSD_Intein-MD3/Intein_ = 0.204, RMSD_Intein-MD3/Intein_ = 0.170) of high confidence (pLDDT_Intein-MD1_= 89.80, pLDDT_Intein-MD2_ = 91.89, pLDDT_Intein-MD3_ = 91.19) and highly confident interaction within the intein complexes (PAE_Intein-MD1_ = 3.02 Å, PAE_Intein-MD2_ = 3.32 Å, PAE_Intein-MD3_ = 2.91 Å) (Figure 3A–I). On another hand, the N-ter and C-ter dystrophin exteins (colored in light pink and light cyan, respectively) in different midi-Dys predicted structures interact differently (Figure 3C,F,I). The midi-Dys 1 PAE showed a highly confident interaction of midi-Dys-5′ and midi-Dys-3′ subunits with 40.74% of pairwise inter-domain PAE less than 10 Å (%PAE_<10Å_) (Figure 3C), and an average contact probability of all highly confident PAE (contact_probs _PAE<10Å_) of 0.00473. Fewer interactions were observed within the prediction of midi-Dys 2 (Figure 3F, %PAE_<10Å_ = 12.12%, contact_probs _PAE<10Å_ = 0.00238) and almost no inter-dystrophin interactions within midi-Dys 3 prediction (Figure 3I, %PAE_<10Å_ = 1.12%, contact_probs _PAE<10Å_ = 0.000625).

Next, we assessed in vitro the reconstitution efficacy of our dual midi-dystrophin constructs. To this end, we performed double transfection with our midi-dystrophin constructs (N-term + C-term) in HEK293T and after 48 h checked the expression of reconstituted midi-Dys (Figure 3J). We performed a capillary Western blot by using two different antibodies for dystrophin, one recognizing the N-terminal part and one recognizing the C-terminal part. We observed the presence of a high protein band (around 240 kDa corresponding to the full size of the Midi-dystrophins) in cells transfected with midi-Dys 1 and midi-Dys 2 with both antibodies, while no band was detected at that position in cells transfected with midi-Dys 3 (Figure 3K). The lower bands detected correspond to the non-reconstituted midi-Dys halves fused to the N- and C-inteins (124.8 kDa and 136.3 kDa for the midi-Dys 1, 141.7 kDa and 138 kDa for the midi-Dys 2, and 138.3 kDa and 131.8 kDa for the midi-Dys 3, respectively). The persistence of these non-reconstituted proteins is probably due to the balance between trans-splicing and protein translation, as shown in other short-term studies on inteins [33]. These results demonstrate a high reconstitution efficiency for midi-Dys 1 and midi-Dys 2, while midi-Dys 3 fails to reconstitute. Importantly, these results indicate that the degree of interaction found within dystrophin domains surrounding intein splits positively correlated with the reconstitution levels (Figure 3J,K). These data revealed that the interactions between the two dystrophin subunits adjacent to the intein splits play an essential role in the protein splicing efficiency.

### 2.4. Restoration of Midi-Dystrophin 1 and Midi-Dystrophin 2 in DBA2-Mdx Mice Following Systemic AAV9 Delivery of Split Midi-Dys

To assess the reconstitution of the midi-dystrophin constructs in vivo, we intravenously injected one month-old DBA2-mdx mice, a severe DMD mouse model, with AAV9-midi-dystrophins at the dose of 2.0 × 10^13^ vg/kg for each vector (Figure 4A). Given that the effectiveness of intein protein trans-splicing can vary between in vitro and in vivo contexts, we included all three midi-dystrophins in our studies, even though midi-Dys 3 did not show efficient reconstitution. In our constructs, the midi-dystrophins cDNA is under the activity of the muscle-specific CK8 promoter, which has been demonstrated to efficiently express transgene in skeletal and cardiac muscles [42]. At 7 weeks post-injection, the viral genome copy number (VGCN) assessment showed 0.29 viral copies per diploid genome for both midi-Dys 1 and midi-Dys 2, while midi-Dys 3 had 0.12 copies per diploid genome in the gastrocnemius, tibialis anterior, and diaphragm muscles (Appendix A). This difference could be attributed to a lack of therapeutic efficacy in midi-Dys 3, leading to the elimination of viral copies within the regenerative myofibers, as previously shown [43]. Immunoblotting data were informative for assessing the ability of the protein trans-splicing of midi-dystrophins through the observation of the reconstituted form over the non-reconstituted one. The capillary Western blot revealed the expression of the reconstituted midi-Dys 1 and midi-Dys 2 (band at 247.1 kDa and 265.6 kDa respectively), while no full-length midi-Dys 3 was observed. Non-reconstituted isoforms of midi-Dys 1 and midi-Dys 3 were observed (bands at 124.8 kDa and 135 kDa, respectively) (Figure 4B). The quantification of reconstituted bands revealed that midi-Dys 1 has higher expression levels compared to midi-Dys 2 (Figure 4C). Protein trans-splicing efficacy was evaluated by calculating the ratio between the full-length midi-Dys and non-reconstituted form by using an antibody that can detect all the midi-Dys constructs. The efficacy of reconstitution observed is 98% for midi-Dys 1 and 2, while midi-Dys 3 has around a 13% efficacy in the gastrocnemius, tibialis anterior, and diaphragm (Figure 4D). Additionally, we observed the robust and uniform expression of midi-Dys 1 and midi-Dys 2 in the histological sections of the diaphragm, while discontinuous localization at the membrane was observed for midi-Dys 3 (Figure 4E). The quantification of dystrophin-positive fibers revealed a higher expression of midi-Dys 1 compared to midi-Dys 2 in the diaphragm and in the tibialis anterior (50–100% of dystrophin positive fibers in mice injected with midi-Dys 1 versus 40–90% in mice injected with midi-Dys 2) (Figure 4F), while no difference was observed in the gastrocnemius. Contrary to the Western blot results, the positive staining of midi-Dys 3 with the N-term antibody may indicate the presence of truncated forms of midi-Dys 3 in the membrane region, as they still contain sarcolemma-binding domain R1-3. All together, these results demonstrated the expression of midi-Dys 1 and midi-Dys 2, with a high reconstitution efficacy and a proper localization of these midi-dystrophins at the membrane.

### 2.5. AAV9 Midi-Dys 1 and Midi-Dys 2 Systemic Delivery in DBA2 Mdx Shows Therapeutic Efficacy

To assess the therapeutic efficacy of the midi-dystrophins, we intravenously injected one-month-old DBA2/mdx mice, with AAV9-midi-dystrophin1, 2, and 3 (2.0 × 10^13^ vg/kg each vector). After 7 weeks from the injection, we measured the biomarkers creatine kinase (CK) and Myomesin 3 (Myom3) in the serum of the mice. The damage to the muscle and the necrosis of the myofibers induce the release of these two molecules in the blood flow, and for this reason, they are currently used as biomarkers for muscular dystrophies [44]. The CK concentration in the serum decreased after gene transfer with midi-Dys 1, while no differences with the mdx control were observed for midi-Dys 2 and 3 (Figure 5A). Consistent with these results, the Myom3 concentration in mice injected with midi-Dys 1 decreases at the same level of the WT, while for midi-Dys 2 and midi-Dys 3, no differences between injected and not injected have been observed (Figure 5B). Then, we assessed the dystrophic pathophysiology in the histological sections of the diaphragm (the most affected muscle in mdx mice) (Figure 5C), gastrocnemius, and tibialis anterior. Fibrosis decreased in mice treated with midi-Dys 1 and to a lower degree with midi-Dys 2, while no decrease was observed in mice injected with midi-Dys 3, in all the three muscles analyzed (Figure 5D). Moreover, the presence of calcific deposits, another hallmark of the dystrophic phenotype, was significantly decreased in mice treated with midi-Dys 1 and midi-Dys 2 (Figure 5E,F). These results were consistent with decreased inflammation and fat infiltration area, which were made visible using the hematoxylin-eosin stainings of gastrocnemius and tibialis anterior of mdx mice injected with midi-Dys 1 and 2 (Appendix A). Gene expression analysis of *Tmem8c*, *TGF-β*, and *Pdgfr-α*—genes linked to the progression of DMD—showed a significant decrease in mice treated with midi-Dys 1 and midi-Dys 2 in both the tibialis anterior and gastrocnemius muscles (Figure 5G). However, treatment with midi-Dys 3 did not reduce the expression of these genes, indicating that it lacks therapeutic effectiveness (Figure 5G). Additionally, a decrease in *Tmem8c*, *TGF-β*, and *Pdgfr-α* expression was observed in the diaphragm of mice injected with midi-Dys 1 (Appendix A). In contrast, no changes in gene expression were seen in mice treated with midi-Dys 2 or midi-Dys 3, which aligns with our previous findings. These results confirm the high therapeutic efficacy of midi-Dys 1 and midi-Dys 2 in restoring key dystrophic features while highlighting the therapeutic inefficacy of midi-Dys 3. Overall, these data show the potential of our strategy and underscore the importance of including functional domains of dystrophin.

## 3. Discussion

In this study, we developed three novel midi-dystrophins using a split-intein dual-AAV strategy, guided using deep learning-based structural predictions to evaluate how design modifications impact protein structure. Due to the limited capacity of AAV vectors, we strategically reduced hinge regions to prioritize the inclusion of functional domains, and we demonstrated that these modifications do not impact the structural stability of the protein. Additionally, we simulated the residue interactions between the two halves of the split midi-Dys proteins, observing strong associations between the N- and C-termini exteins in midi-Dys 1 and 2, with a weaker interaction in midi-Dys 3. The strong residue interaction in the adjacent sites of the intein correlates with efficient protein trans-splicing (PTS) both in vitro and in vivo in DBA2-mdx mice. Consistent with these findings, we demonstrated the high therapeutic efficacy of midi-Dys 1 and 2 in DBA2-mdx mice, underscoring their potential therapeutic applications.

AAVs are the most commonly used vectors for gene therapy due to their non-integrative nature, lower immune response compared to adenoviruses and lentiviruses, and their ability to be engineered for specific muscle targeting. However, gene replacement for DMD remains challenging with AAV vectors due to their limited cargo capacity. The most promising approach involves delivering micro-dystrophin (µDys) to restore the mechanical properties of myofibers by including membrane and cytoskeleton binding domains. However, limited clinical trial improvements suggest µDys may be sub-optimal in humans, potentially due to missing key protein domains. This suggests that expressing larger dystrophin proteins may be necessary for more effective disease correction. In our approach, we incorporated additional functional domains that are absent in micro-dystrophin. Specifically, we added interaction regions with the membrane, the cytoskeleton, Par1b, and nNOS. The connection between dystrophin and nNOS, through the R16-R17 region, is well-known for its importance. In fact, the delivery of µDys including the R16-R17 has been shown to improve muscle strength and performance compared to those without the nNOS binding ability [45,46]. Additionally, the binding of dystrophin with Par1b was shown to be crucial for the proper asymmetrical division of satellite cells, which is vital for effective regeneration in dystrophic conditions [11]. In view of these added functional domains, it may be necessary to exacerbate the pathological condition to fully assess their impact, such as through cardiotoxin-induced muscle damage or long-term experiments, where µDys alone is ineffective [18].

In our study, in order to accommodate these additional domains, we strategically removed a great part of the hinge 1 by reducing it by 56.25%, maintaining a good therapeutic efficacy at low dose. These results are consistent with the recent work of Duan’s lab, where they demonstrated that hinge 1 is not necessary for the activity of micro-dystrophin in ameliorating the DMD phenotype, whereas hinge 4 is crucial for improving the pathophysiology [47]. Interestingly, the region that encodes for hinge 1 corresponds to an epitope linked with an immune response against dystrophin’s exon 8–11 in µDys injected patients. Indeed, a recent study reported that five DMD patients across four different clinical trials by Sarepta, Roche (using Sarepta’s vector), Pfizer, and Genethon—receiving three different gene therapy products varying in AAV serotype, promoter, and dose—experienced strikingly similar severe adverse events suggestive of a cytotoxic T-cell immune response against micro-dystrophin proteins [48]. To the best of our knowledge, no other study has demonstrated the therapeutic efficacy of hinge-reduced midi- or micro-dystrophin, proving the impact of our study for the development of less-immunogenic DMD gene therapies. Further efforts will focus on investigating potential immune responses in comparison to micro-dystrophin to understand the impact of the hinge 1 reduction in developing an immune response.

To accommodate additional functional domains in our midi-Dys constructs, we harnessed the split-intein technology to deliver dual-AAVs. Intein activity is known to be context-dependent, particularly at the level of the amino-acid environment surrounding their ligation junctions (N- and C-exteins). For GP41-1, the deletion of the native extein residues was shown not to impact the cleavage reaction yields [39], allowing the insertion of the intein into sequences without adding extra residues to the targeted proteins. Since trans-splicing occurs at the protein level, the design of the split proteins (N- and C-terminal) must be precise to avoid incorrect polypeptide folding and to ensure efficient PTS. Some half-peptides may exhibit limited stability or incorrect targeting within the cell, resulting in low recombination efficiency between the two proteins. To evaluate the structural stability of the midi-Dys half-peptides and the residue interactions involving the intein, we used AlphaFold3, a deep learning algorithm that predicts protein structures from amino acid sequences. A major advantage of AlphaFold is its high accuracy in predicting protein conformations, including engineered forms that do not exist in nature, such as new therapeutic proteins. Our analysis confirmed the correct folding of our constructs and demonstrated the stability of the intein within the protein sequence. Additionally, AlphaFold3’s capability to predict protein–protein interactions within biomolecular complexes allowed us to observe residue interactions between intein-adjacent sites in some constructs, correlating with efficient PTS in vitro. This approach enabled the development of two novel and efficient midi-dystrophin constructs. Upcoming preclinical studies will focus on assessing the advantages of this method compared to existing approaches, as well as optimizing safety and efficacy parameters.

In conclusion, we validated the efficacy of our in silico approach, which predicts inter-domains interactions, and we confirmed our structural hypothesis on the importance of extein interactions for the occurrence of protein trans-splicing. Overall, our findings underscore the remarkable potential of deep learning-based tools in the design of novel gene therapy products, paving the way for even more advanced approaches in the treatment of genetic diseases.

## 4. Material and Methods

### 4.1. Midi-Dystrophins Design and Viral Particle Preparation

Midi-dystrophin 1, midi-dystrophin 2, and midi-dystrophin 3 protein sequences were derived by dystrophin muscular isoform (Dp427m; RefSeq: NM_004006.3) and optimized to remove the rare human codons, CpG, and possible alternative ORFs via GeneArt service (Life Technologies, Carlsbad, CA, USA). Gene fragments were produced through GeneArt gene synthesis (Life Technologies) and cloned into a plasmid containing CK8 promoter and poly A SV40. Plasmids were amplified by endotoxin-free methods. For recombinant AAV production, HEK293-T cells (obtained from Stanford University School of Medicine), cultured in suspension, were transfected with the three plasmids coding for the adenovirus helper proteins, the AAV Rep and Cap proteins, and the ITR-flanked transgene expression cassette. Three days after transfection, cells were harvested, chemically lysed, and treated with benzonase (Merck-Millipore, Darmstadt, Germany). After filtration, viral capsids were purified using affinity chromatography, formulated in sterile PBS, and the vector stocks were stored at −80 °C. The titers of AAV vector were determined by using digital droplet polymerase chain reaction (ddPCR). Viral particles were treated with DNAse I for 30 min at 37 °C (Invitrogen, Waltham, MA, USA) and then viral DNA was amplified by using polyA SV40 specific primers with ddPCR Supermix for probes (Biorad, Hercules, CA, USA).

### 4.2. Structural Prediction Using AlphaFold3

The predicted 3D prediction of the midi-dystrophins structure was obtained with AlphaFold3 server (AlphaFold Server, https://alphafoldserver.com/about (accessed on 22 May 2024)). The midi-dystrophin proteins were trimmed in three parts: the N-terminal region (including the ABD1, the hinge 1, and R1), central region of the rod domain, and the C-terminal region (R24, hinge 4, the CR, and the C-term). The first and the last parts are share commonalities in the three midi-Dys, while the central rod domain was predicted separately for all the proteins. For protein subunits interaction, prediction was obtained using AlphaFold3 and root square mean deviation (RMSD) was defined using Pymol (Molecular Graphics System, Version 1.3, Schrödinger, LLC). The predicted local distance difference test (pLDDT) was carried out and predicted aligned error (PAE) was calculated using a custom Python code. The amino-acid sequences of midi-Dys, as well as the original Python code, are available on GitHub (GitHub—GNT-DDC/midi-dystrophin-af3: in silico structural prediction for the generation of novel performant midi-dystrophins based on intein-mediated dual AAV approach, https://github.com/GNT-DDC/midi-dystrophin-af3 (released on 22 September 2024)).

### 4.3. In Vitro Assessment of Protein Trans-Splicing

The in vitro assessment of protein trans-splicing was performed via the transfection of split-intein GFP and midi-dystrophins plasmid. The GFP sequence was obtained using NIH Uniprot database (GenBank: QAA95706.1). Its coding sequence was optimized to remove rare human codons and possible alternative ORFs using GeneArt service (Life Technologies). The GFP sequence was then split into two parts, whereas a Serine was present in position +1 in the C-terminal extein. Additionally, a sequence HA tag was added at the end of the N-term vector. GFP gene fragments were produced using GeneArt gene synthesis (Life Technologies) and cloned into a plasmid containing CK8 promoter and poly A SV40. HEK293T was transfected at 70% of confluence using a Lipofectamine 3000 kit (Invitrogen, MA, USA) accordingly to the manufacturer’s instructions, with either a GFP or midi-Dys N- and C-terminal vector. After 72 h, cells were harvested and washed twice with PBS. The flow cytometry experiment was performed using the CytoFLEX S (Beckman-Coulter, Brea, CA, USA) and FloJO software v10.9 (TreeStar, Ashland, OR, USA).

### 4.4. Capillary Western Blot and Protein Trans-Splicing Efficacy Assessment

The capillary Western blot of midi-dystrophins was performed on a Simple Western™ Jess system (ProteinSimple, Bio-Techne, Minneapolis, MN, USA) according to the manufacturer’s instructions, using a 60–440 kDa separation module (ProteinSimple SMW004) and the anti-mouse detection module (Protein simple DM-002) or anti-rabbit detection module. In brief, protein samples consisting of either green fluorescent protein (GFP) or midi-dystrophin proteins obtained from lysed HEK 293-transfected cells were loaded at a concentration of 0.5 µg/µL. For capillary Western blot on muscle lysates, proteins were extracted in RIPA buffer and 2.5 ng was used for dystrophin detection in mdx mice injected with vehicle, AAV-midi-dystrophins, or AAV-micro-dystrophin. Primary antibodies used for recognizing the N-term and the C-term part of dystrophin are listed in Table 1. PTS efficiency was calculated through conducting the ratio between the non-reconstituted and reconstituted protein fragments for each midi-dystrophin construct.

### 4.5. Animal Care and Use

DBA2 (B6;129S4-DBA2tm1Cpr/J, strain #000671) and DBA2-mdx (D2.B10-Dmdmdx/J) mice were supplied by the Jackson Laboratory. All animal procedures were approved by the National Ethical Committee, C2EA-51 (Evry-Courcouronnes, France), and the French Ministry of Research (MESRI) and received a national agreement number (APAFiS #38006). Four-week-old Dba2_mdx and Dba2_WT male mice were injected via retro-orbital injection with dual midi-dystrophin AAV9 vectors at a dose of 2 × 10^13^ vg/kg for each vector (total dose of 4 × 10^13^ vg/kg) and with a micro-dystrophin AAV9 vector at the dose of 2 × 10^13^ vg/kg. AAV9-injected mice were compared to Dba2_mdx mice injected with PBS (negative control) as well as to the Dba2 WT mice (positive control). Seven weeks post-injection, an escape test was performed (see below), and muscles and serum were collected for molecular and histological analysis.

### 4.6. Genomic DNA Extraction and Viral Copy Number Analysis

Genomic DNA was extracted from the muscles using a NucleoMag Pathogen kit (Macherey Nagel, Allentown, PA, USA) according to the manufacturer’s instructions, and was purified using the KingFisher Flex purification system (Thermo-Fisher-Scientific, Waltham, MA, USA). Droplet digital PCR was performed for the detection of vector genome number by using N-terminal and C-terminal specific primers. For the detection of the N-terminal midi-dystrophins and micro-Dys (at the 5′ end), R1-specific primers were used. For the detection of the C-terminal midi-dystrophin and micro-Dys (at the 3′ end), R24-specific primers were used in combination with ddPCR for probes technology (Biorad, Hercules, CA, USA). The results were normalized for the titin genomic region and were shown as the viral copy number (VCN) per diploid genome. The primers and probes details are shown in Table 2.

### 4.7. Histological Staining

Skeletal muscles (tibialis anterior, diaphragm, and gastrocnemius) were sampled and frozen in isopentane cooled in liquid nitrogen. Transverse cryosections (8–10 µm) were prepared from frozen muscles, air-dried, and stored at −80 °C. Muscle sections were processed for Sirius red, red Alizarin, and hematoxylin-eosin stainings. The sections were visualized on an Axioscan Z1 automated slide scanner (Carl Zeiss, Oberkochen, Germany) using a Plan APO 10X/0.45 NA objective. The sections were immunostained overnight at 4 °C; the primary antibodies are listed in Table 1. Following three washes with PBS, the muscle sections were incubated for 1 h at room temperature with a goat secondary antibody conjugated with Alexa Fluor 488 or 594 dye (Invitrogen, Carlsbad, CA, USA dilution 1:1000). The sections were then mounted using DAPI-Fluoromount-G (Southern Biotech, Birmingham, AL, USA) and visualized either on a LEICA TCS-SP8 confocal microscope (Leica, Wetzlar, Germany) with a 63X APO CS2 1.4 NA objective or on an Axioscan Z1 automated slide scanner (Zeiss) with a Plan APO 10X/0.45 NA objective.

### 4.8. Fibrosis and Calcific Quantification

Sirius red and Alizarin-stained transverse sections were used for fibrosis and calcification assessment using an open-source software for bioimages analysis QuPath (version 0.3.2). Two small artificial neural networks were trained on three representative muscle sections to classify positive and negative pixels. The first one is for delimiting the tissue to be considered and the second one is for identifying the region of collagen deposits or calcified fibers to be quantified. We subsequently used these networks for each muscle scan to quantify fibrotic and calcified surface areas. The results were expressed in percentage (%) of areas in relation to the total muscle section surface area.

### 4.9. Dystrophin Positive Fibers Quantification

Muscle sections were stained with laminin to label cell membranes. The Cellpose2 cyto2 model 10 was fine-tuned using manually labeled images of myofibers based on laminin staining. This fine-tuning was performed with hyperparameters set to 200 epochs, a learning rate of 0.05, and a weight decay of 0.0001. The labeled dataset was carefully prepared to allow the model to simultaneously segment myofibers while ignoring areas with low-quality staining.

Once fine-tuned, these models were utilized to extract myofiber masks. The reconstruction of myofiber masks from whole-scan images was performed using the Cellpose package 10. These reconstructed masks were then converted into Regions of Interest (ROIs), with each ROI corresponding to an individual myofiber, using the Labels_To_Rois.py FIJI plugin 11.

Muscle sections were co-labeled with laminin for membrane labeling and dystrophin for further analysis. As with the initial labeling, ROIs were generated based on membrane labeling and used for the subsequent quantification of dystrophin signals using a FIJI macro.

### 4.10. Serum Biomarkers Quantification

Blood samples were collected using retro-orbital bleeding and quickly centrifuged for 10 min at 8000 rpm. Sera were harvested and stored at −80 °C until measurement. For serum MYOM3 quantification, a custom sandwich ELISA assay was employed. Initially, a polyclonal MYOM3 antibody (Proteintech, 17692-1-AP), diluted 1/10, was coated onto a 96-well plate and incubated overnight at 4 °C. After this incubation, the plate was washed three times with PBST and then blocked with a saturation solution of 3% BSA in PBS. Dilutions of the serum samples were then added to the plate and incubated for 2 h at room temperature. For detection, a custom monoclonal antibody (Proteogenix, REF: 51-H1-B4) coupled with SULFO-TAG (MSD) was used, and the plate was incubated for an additional 2 h at room temperature. After incubation, the plate was washed three times with wash buffer (0.05% Tween-20 in PBS). The absorbance of the SULFO-TAG was measured using the MESO Quickplex SQ 120 (MSD). To quantify the MYOM3 concentration in the serum samples, a set of concentration-defined MYOM3 peptides (His-tagged, Proteogenix) was included in the same experiment and served as the standard for calculation. For creatin phosphokinase (PK) quantification, 10 μL of mouse serum was used to calorimetrically measure creatine phosphokinase concentration using FUJI DRI-CHEM nx500 system (DMV Imaging).

### 4.11. RNA Extraction and Gene Expression Analysis

Total RNA was extracted from frozen muscle tissue using an IDEAL32 extraction robot. Any DNA contamination in the RNA samples was removed with the TURBO DNA-free kit (Thermo Fisher Scientific). For gene expression analysis, 1000 ng of total RNA was reverse-transcribed using a mix of random oligonucleotides and oligo-dT, along with the RevertAid H Minus First Strand cDNA Synthesis Kit (Thermo Fisher Scientific). The resulting cDNA was then quantified using qPCR on the LightCycler 480 system (Roche) using Taqman Gene Expression Assays (Thermo Fisher Scientific). Each PCR reaction was conducted in duplicate, and the data were normalized across samples using Rplp0 as a reference gene.

### 4.12. Statistical Analysis

All data were analyzed using GraphPad Prism 9.5.1 software. Parametric tests such as *t*-tests and ANOVA were used for statistical comparison. To compare the two groups, initially, the F-test was used to compare variances. If there was no difference in variances, a statistical comparison was performed using an unpaired *t*-test. To compare multiple groups, we used one-way ANOVA with Tukey’s correction for multiple comparison tests. For grouped analysis, we performed a two-way ANOVA statistic test with multiple comparisons with the main row effect. Results were considered significantly different at *p* < 0.05. Graphs were generated using Graphpad Prism v9 or R version 3.6.2. The figures display the mean ± standard deviation.

## Figures and Tables

**Figure 1 ijms-25-10444-f001:**
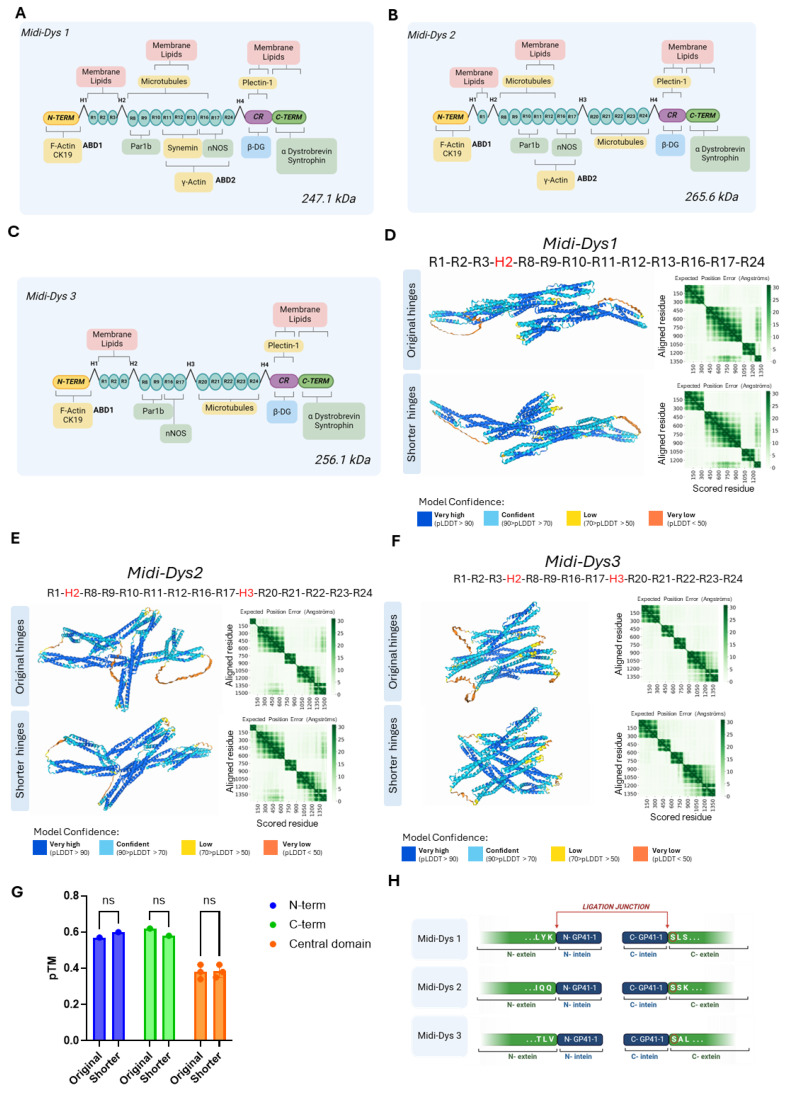
Rational design of novel midi-dystrophin. (**A**) Schematic representation of midi-dystrophin 1 (midi-Dys 1) which has a molecular weight of 247.1 kDa. (**B**) Schematic representation of midi-dystrophin 2 (midi-Dys 2) with its molecular weight of 265.6 kDa. (**C**) Scheme representing the domains included in midi-dystrophin 3 (midi-Dys 3) and its molecular weight of 256.1 kDa. For (**A**–**C**): In yellow, the binding domain with cytoskeletal protein is represented, in pink, the binding with sarcolemma lipids, in green, the binding of signaling proteins, and in blue, the binding of transmembrane proteins (β-DG = β dystroglycan). (**D**) The 3D structure of the central rod domain of midi-Dys 1 including the R1-3, hinge 2, R8-13, R16-17, and R24 with the original dystrophin hinge and after the shortening. (**E**) The 3D structure of the central rod domain of midi-Dys 2 including the R1, hinge 2, R8-12, R16-17, hinge 3, and R20-24 with the original dystrophin hinge and after the shortening. (**F**) The 3D structure of the central rod domain of midi-Dys 3 including the R1-3, hinge 2, R8-19, R16-17, hinge 3, and R20-24 with the original dystrophin hinge and after the shortening. For (**D**–**F**): the green heatmaps represent the expected position error (if 0 is green and if 30 is white) for each aligned residue. The corresponding color of each portion of the protein indicates the level of confidence in the prediction: dark blue corresponds to high confidence in the prediction, light blue corresponds to medium confidence, yellow corresponds to low confidence, and orange corresponds to very low confidence in the prediction. (**G**) The predicted template modeling (pTM) score for structural predictions of midi-Dys 1, 2 and 3 central rod domains for shorter and original. (**H**) The insertion sites of GP41-1 split-intein, highlighting the ligation junction and the extein flanking amino acids. Created with BioRender.com.

**Figure 2 ijms-25-10444-f002:**
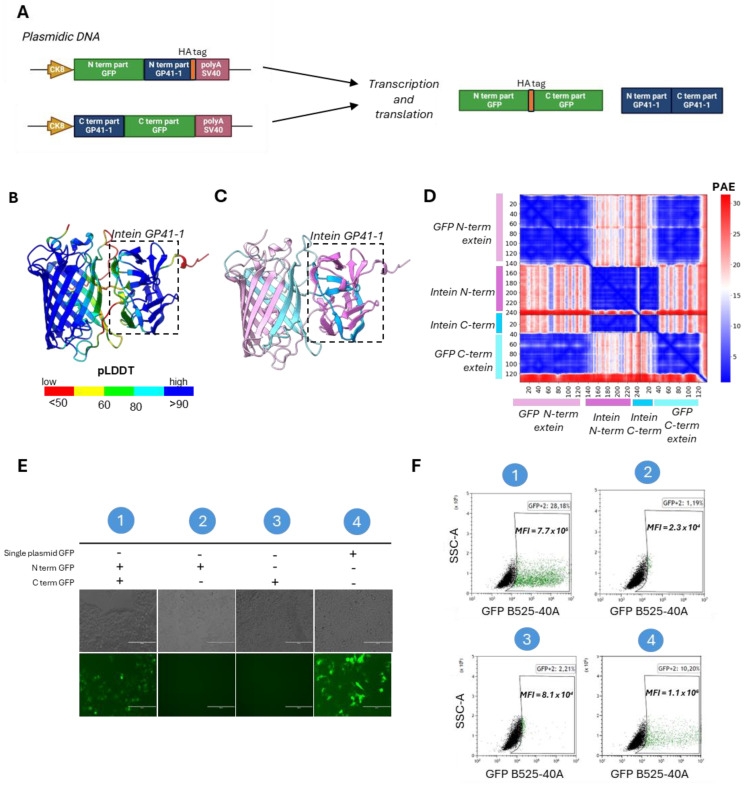
Deep learning-based predictions of intein residue interactions and in vitro validation of GFP intein-split PTS. (**A**) Schematic representation of the dual plasmid GFP GP41-1 split-intein, followed by a HA tag. (**B**) The predicted structure of GP41-1 intein and GFP colored by the predicted local distance difference test (pLDDT) (red, 0–50; yellow, 50–60; green, 60–80; cyan, 80–90 and blue, more than 90). (**C**) The same prediction colored by the structured protein chain. GP41-1 intein is highlighted by a square box. Light pink: −200aa region; pink: N-term GP41-1 intein; blue: C-term intein; light blue: +200aa region. (**D**) PAE matrix highlighting the predicted error of interaction between the intermolecular domains. (**E**) Fluorescent images of HEK293T lipofected with single GFP plasmid (containing the GFP in one plasmid sequence) or with N-terminal GFP plasmid and C-terminal GFP plasmid. Single vector controls are included. Condition 1: N-term GFP + C-term GFP. Condition 2: N-term GFP alone. Condition 3: C-term GFP alone. Condition 4: Single plasmid GFP alone. (**F**) FACS analysis of HEK293T lipofected with GFP GP41-1 split-intein dual plasmid. GFP mean fluorescent intensity (MFI) over GFP-positive cells is indicated in each experimental condition. Condition 1: N-term GFP + C-term GFP. Condition 2: N-term GFP alone. Condition 3: C-term GFP alone. Condition 4: Single plasmid GFP alone. Created with BioRender.com.

**Figure 3 ijms-25-10444-f003:**
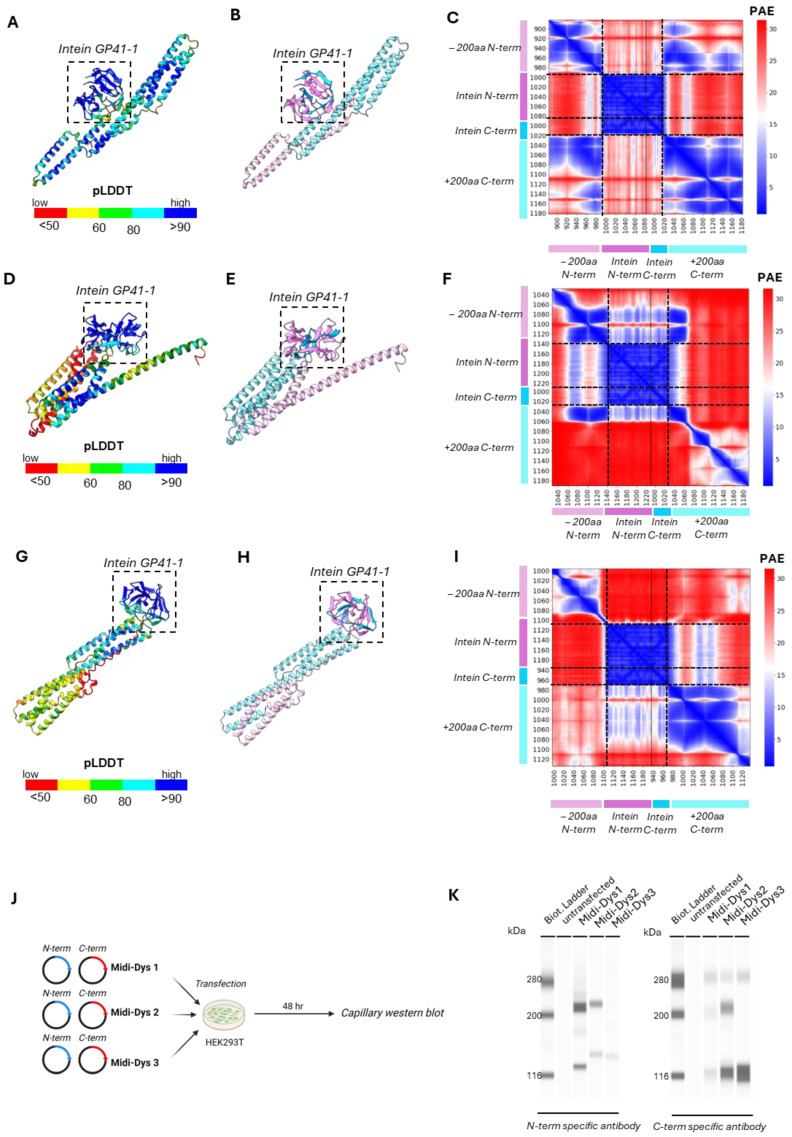
In silico predictions of intein residue interactions and validation of midi-Dys intein PTS. (**A**) Predicted structure of GP41-1 intein and midi-Dys 1 200aa around the split sites colored by pLDDT. (**B**) The same prediction colored by a structured protein chain. GP41-1 intein is highlighted by a square box. (**C**) PAE matrix highlighting the predicted error of interaction between the intermolecular domains. (**D**) Predicted structure of GP41-1 intein and midi-Dys 2 200aa around the split sites. (**E**) The same prediction colored by chain. (**F**) PAE matrix highlighting the predicted error of interaction between the N-terminal chain and the C-terminal chain. (**G**) Predicted structure of GP41-1 intein and midi-Dys 3 200aa around the split. (**H**) The same prediction colored by chain sites. For (**A**,**D**,**G**): the structures are colored by pLDDT (red, 0–50; yellow, 50–60; green, 60–80; cyan, 80–90 and blue, more than 90). For (**B**,**E**,**H**): GP41-1 intein is inside a square box. Light pink: −200aa region; pink: N-term GP41-1 intein; blue: C-term intein; light blue: +200aa region. (**I**) PAE matrix highlighting the predicted error of interaction between the two split chains. (**J**) Schematic representations of midi-dystrophins lipofection in HEK293T and capillary Western blot. Created with BioRender.com. (**K**) Capillary Western blot of HEK293T lipofected with midi-Dys 1-2-3 with N-term and C-term specific antibody. Specific bands are visible at 280 kDa. Reconstituted midi-Dys 1-2 are visible around 220–230 kDa, while non-reconstituted midi-Dys 1-2-3 are visible around 120–130 kDa.

**Figure 4 ijms-25-10444-f004:**
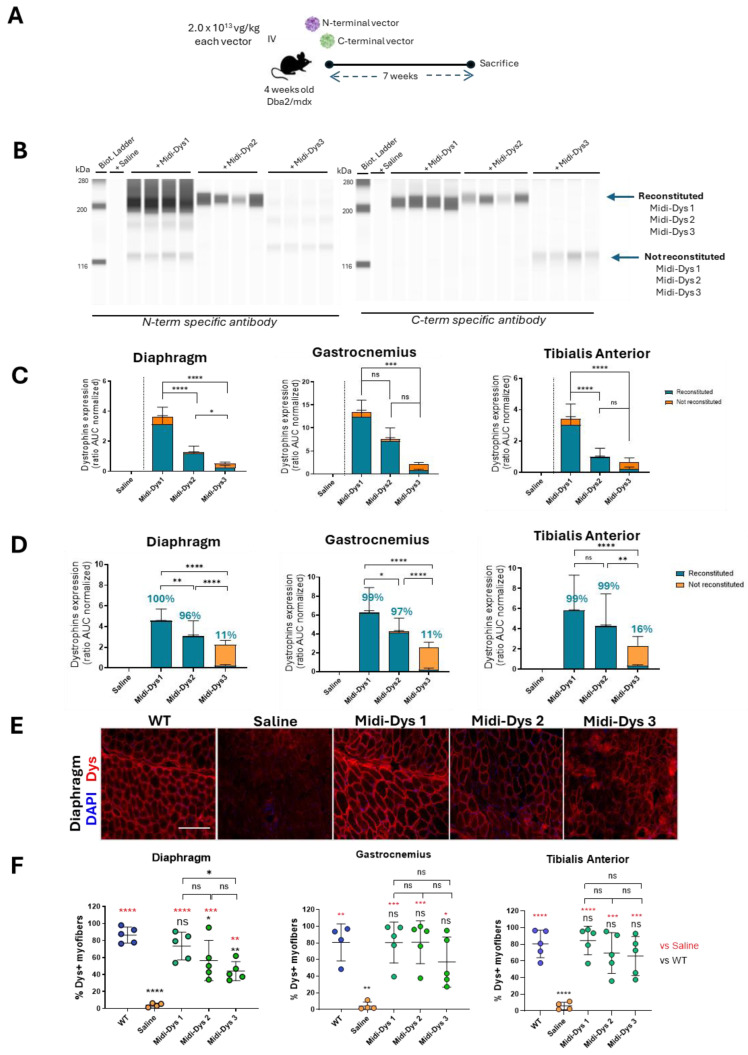
Efficacy of reconstitution of midi-dystrophins in Dba2-mdx mouse model. (**A**) Schematic representation of the in vivo protocol. One month old Dba2/mdx mice were injected with midi-Dys 1, midi-Dys 2, and midi-Dys 3 at a dose of 2.0 × 10^13^ vg/kg each vector (total dose 4.0 × 10^13^ vg/kg). Tissues were collected after 7 weeks from the intravenous injection. (**B**) The capillary Western blot of protein lysates of diaphragm at 7 weeks post-injection in Dba2-mdx mice with N-term and C-term specific antibodies. Reconstituted midi-Dys are visible around 200 kDa, while non-reconstituted midi-Dys are visible around 116 kDa. (**C**) TheQuantification of midi-Dys 1-2-3 and micro-Dys protein expression after capillary Western blot using N-term specific antibody in gastrocnemius, tibialis anterior, and diaphragm at 7 weeks post-injection in Dba2-mdx mice. Values are represented as normalized area under the curve (AUC). (**D**) The quantification of midi-Dys 1-2-3 (reconstituted and non-reconstituted) protein expression after capillary Western blot using C-term midi-Dys specific antibody in gastrocnemius, tibialis anterior, and diaphragm at 7 weeks post-injection in Dba2-mdx mice. Values are represented as normalized area under the curve (AUC). Blue values represent the percentage of reconstitution, calculated as the protein expression of reconstituted midi-Dys over the total. (**E**) Representative dystrophin IHF pictures of the diaphragm at 7 weeks post-injection in Dba2-mdx mice using N-term specific antibody. Scale bar = 100 µm (**F**) The quantification of dystrophin positive myofibers in gastrocnemius, tibialis anterior, and diaphragm at 7 weeks post-injection in Dba2-mdx mice. The percentage of dystrophin + fibers is represented of number of dystrophin positive fibers over fibers positive for laminin. For all the panels: *N* = 4–5. One-way ANOVA statistic test. * *p* < 0.05, ** *p* < 0.005, *** *p* < 0.001, **** *p* < 0.0001, ns = not significant.

**Figure 5 ijms-25-10444-f005:**
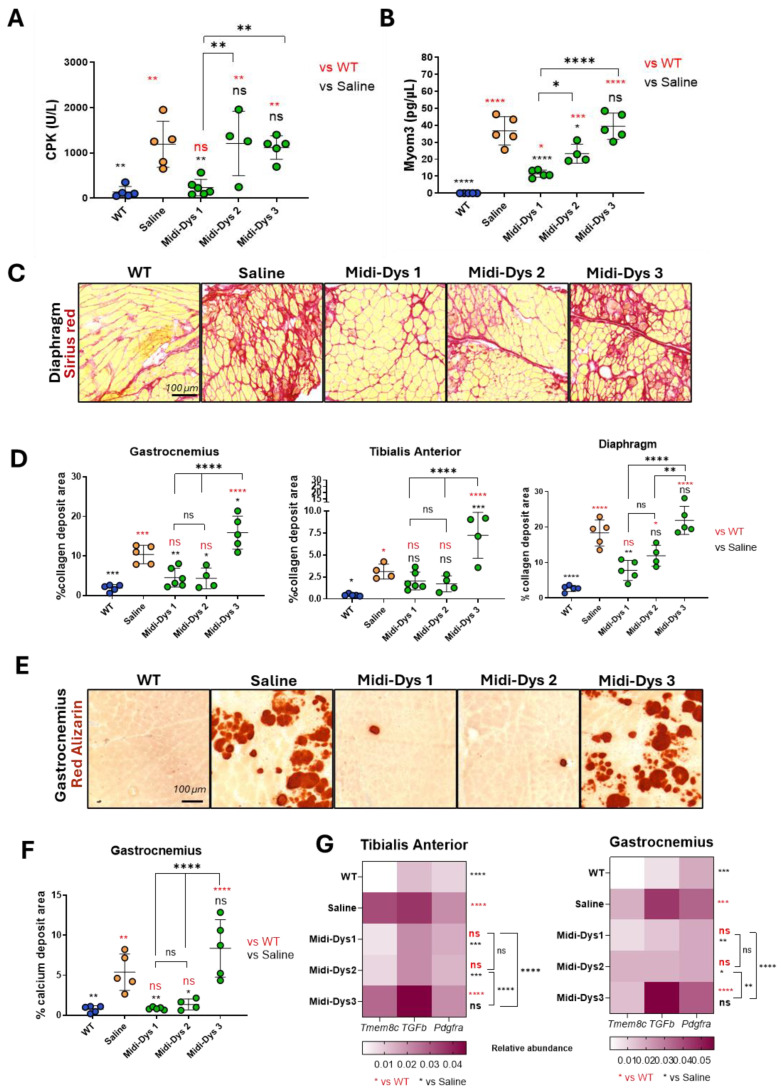
Functional and histopathological therapeutic efficacy in Dba2-mdx mice. (**A**) CPK (creatine phosphokinase) concentration in the sera of Dba2-mdx mice injected with midi-Dys 1-2-3. Results are shown as concentration defined units/liter (U/L). (**B**) Myom3 (Myomesin 3) concentration in the sera of Dba2-mdx mice injected with midi-Dys 1-2-3. Results are represented as Myom3 concentrations defined as pg/µL. (**C**) Representative images of Sirius red coloration in the diaphragm of Dba2-mdx mice 7 weeks post-injection. Red-colored regions are highly fibrotic and contain collagen deposits. (**D**) The quantification of collagen deposits area in gastrocnemius, tibialis anterior, and diaphragm at 7 weeks post-injection in Dba2-mdx mice. The results are shown as the percentage of collagen deposits area over the total area of the muscle cut. (**E**) Representative images of red Alizarin coloration in the gastrocnemius of Dba2-mdx mice 7 weeks post-injection. Red areas are indicated as a region containing calcium deposits. (**F**) The quantification of the calcium deposits area in the gastrocnemius of Dba-mdx injected mice. The results are shown as the percentage of calcium positive regions over the total area of the muscle cut. (**G**) Gene expression analysis of *Tmem8c, TGF-β*, and *Pdgfr-α* in tibialis anterior and gastrocnemius of WT mice and DBA2/mdx mice injected with saline, midi-Dys 1, midi-Dys 2, or midi-Dys 3. Results are shown as a heatmap of relative abundance of gene expression over P0 normalizer gene. For all the panels: *N* = 4–5. For panel (**A**–**F**): One-way ANOVA statistic test. * *p* < 0.05, ** *p* < 0.005, *** *p* < 0.001, **** *p* < 0.0001, ns = not significant. For panel (**G**): Two-way ANOVA statistic test and multiple comparisons with main row effect. * *p* < 0.05, ** *p* < 0.005, *** *p* < 0.001, **** *p* < 0.0001, ns = not significant.

**Table 1 ijms-25-10444-t001:** List of commercial antibodies used for capillary Western blot, immunofluorescence, and Myom-3 ELISA.

Antigene	Epitope	Ref	Diluition		
			Capillary Western Blot	Immunofluore-Scence	ELISA
Dystrophin	N-term specific antibody	Leica Cat#NCL-DYSB, RRID:AB_563691	1/20	1/10	/
	C-term specific antibody	DSHB Cat#MANCHO11(9F2), RRID:AB_2618131	1/100	/	/
Laminin	Polyclonal	Thermo Fisher Scientific Cat# PA5-115490, RRID:AB_2900126	/	1/100	/
Myomesin-3	Polyclonal	Proteintech Cat#17692-1-AP, RRID:AB_2146624	/	/	1/100

**Table 2 ijms-25-10444-t002:** List of primers used for viral genome copy number (VGCN) in ddPCR.

Amplified Region	Name	Sequence (5′->3′)	Probe	Target Vector
Spectrin-like region 1	R1_Forward	CCAGGGAGAGATCAGCAATG	/	Midi-Dys 1 Midi-Dys 2 Midi-Dys 3
	R1_Reverse	GGCTGTTAGGTCCATCATGTAG	/	
	R1_Probe	GGCTGTTAGGTCCATCATGTAG	FAM	
Spectrin-like region 24	R24_Forward	ACAGATGAGCTGGACCTGAA	/	Midi-Dys 1 Midi-Dys 2 Midi-Dys 3
	R24_Reverse	TGGTCCTGTAGGCTGTCAAT	/	
	R24_Probe	CTGAGGCAGGCTGAGGTGATCAAG	VIC	
Mouse Titin exon 5	TitinMex5_Forward	TTCAGTCATGCTGCTAGCGC	/	genomic DNA
	TitinMex5_Reverse	AAAACGAGCAGTGACGTGAGC	/	
	TitinMex5_Probe	TGCACGGAAGCGTCTCGTCTCAGTC	5Cy5	

## Data Availability

All data associated with this study are present in the paper or the Appendix A. Protein sequences and original Python codes are available on GitHub (Palmieri, L (2024) In silico structural prediction for the generation of novel performant Midi-dystrophins based on intein-mediated dual AAV approach. GitHub: https://github.com/GNT-DDC/midi-dys-af3, accessed on 22 May 2024).

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
