# Peer review of "In Silico Structural Prediction for the Generation of Novel Performant Midi-Dystrophins Based on Intein-Mediated Dual AAV Approach"

_ijms, 2024, doi:10.3390/ijms251910444_

Round 1

Reviewer 1 Report

Comments and Suggestions for Authors

Palmieri et al. developed three novel midi-dystrophins (midi-dys), guided by deep learning-based tools, and using a split-intein dual-AAV strategy, as therapeutic alternatives to the currently available micro-dystrophin in Duchenne Muscular Dystrophy (DMD).

Of the three midi-dystrophins, two show promising results in vitro (transfection in HEK293T cells proves reconstitution of midi-dys, and GFP reporter system established high protein trans-splicing efficiency), and in vivo (using DBA2/mdx mice, to evaluate protein trans-splicing efficiency, and therapeutic efficiency, supported by decrease in muscle fibrosis and calcium deposits by histology, but also decrease in serum biomarkers of muscle damage).

Overall, the work presented is robust and shows the potential of deep-learning methods to aid in the optimization of constructs for therapeutic purposes. Presented findings match existing evidence, which partially reduces their novelty,  of hinge-reduced midi-dys therapeutic efficacy (Wasala et al., 2023, cited in the paper), and by a recent study using a split-intein approach, also through GP41-1 (Tasfaout et al., 2024, https://doi.org/10.1038/s41586-024-07710-8).

Minor concerns

-       Fig. 3K: What are the bands at 280 for all three midi-dys? Also, modify figure legend moving the BioRender mention to 3J

-       Fig. 4 B: what is the point of having the column of the WT dys if it cannot be detected (also probably would be at a higher molecular weight if present)? What does the smear visible (between 230 nd 280) especially for midi-dys1 represent?

-       Fig. 4 C and D: keep color and text legend consistent between figures 

-       Fig. 5: To complement fibrosis and calcium deposits characterization in mdx, it would be fair to stain for lipid deposits (Oil Red O)

-       Line 208 typo: interact (with) each other

Author Response

Comment 1: 

Palmieri et al. developed three novel midi-dystrophins (midi-dys), guided by deep learning-based tools, and using a split-intein dual-AAV strategy, as therapeutic alternatives to the currently available micro-dystrophin in Duchenne Muscular Dystrophy (DMD).

Of the three midi-dystrophins, two show promising results in vitro (transfection in HEK293T cells proves reconstitution of midi-dys, and GFP reporter system established high protein trans-splicing efficiency), and in vivo (using DBA2/mdx mice, to evaluate protein trans-splicing efficiency, and therapeutic efficiency, supported by decrease in muscle fibrosis and calcium deposits by histology, but also decrease in serum biomarkers of muscle damage).

Overall, the work presented is robust and shows the potential of deep-learning methods to aid in the optimization of constructs for therapeutic purposes. Presented findings match existing evidence, which partially reduces their novelty, of hinge-reduced midi-dys therapeutic efficacy (Wasala et al., 2023, cited in the paper), and by a recent study using a split-intein approach, also through GP41-1 (Tasfaout et al., 2024, https://doi.org/10.1038/s41586-024-07710-8).

Response 1: 

We thank the reviewer for the comments raised that helped us to improve the quality and clarity of the manuscript. Please, find a point-to-point response.

Comment 2: Fig. 3K: What are the bands at 280 for all three midi-dys? Also, modify figure legend moving the BioRender mention to 3J

Response 2: The bands at 280 kDa in the Capillary western blot with C-terminal antibody is a non-specific signal of the antibody since they are not visible with the N-Terminal antibody. We added the info about the non-specific band at line 292. Biorender citation moved.

Comment 3: Fig. 4 B: what is the point of having the column of the WT dys if it cannot be detected (also probably would be at a higher molecular weight if present)? What does the smear visible (between 230 nd 280) especially for midi-dys1 represent?

Response 3: we agree with the reviewer that the inclusion of the Control mice in this case was useless and we removed the WT mice control. The smear is due to the high concentration of Midi-Dys1 compared to Midi-Dys2. In fact, to visualize the two proteins represented, we have a strong signal of Midi-Dys1, which seems a smear.  However, looking at the graph of the curve of the capillary western blot, we can see only one strong peak, that corresponds to the Midi-Dys1 reconstituted, and one just before the curve (Figure available if requested). Nevertheless, this is just visible with the N Terminal antibody, so we can exclude that is a longer form of dystrophin.

Comment 4: Fig. 4 C and D: keep color and text legend consistent between figures 

Response 4: We modified the legend color and text accordingly.

Comment 5: Fig. 5: To complement fibrosis and calcium deposits characterization in mdx, it would be fair to stain for lipid deposits (Oil Red O)

Response 5: We agree with the reviewer on this important parameter of DMD pathology.  For this purpose, we included hematoxylin-eosin staining representative pictures indicative of the inflammation and fat infiltration. Although the coloration is not specific for lipids, the histological analysis shows clearly the increased fat deposits in the DMD condition and its reduction following Midi-Dys1 and 2 treatments. New results are shown in the new figure panels S3C and described in line 374.

Comment 6: Line 208 typo: interact (with) each other

Response 6: We fixed that.

Reviewer 2 Report

Comments and Suggestions for Authors

The article is an original and interesting work demonstrating how using intein technologies and structural modeling with AlphaFoldF3 it is possible to create new therapeutically competent protein hybrids. The object of the study was the protein dystrophin, the absence of which leads to severe muscular dystrophies, which can be partially compensated by viral delivery of functionally important domains of dystrophin, called µdystrophins. The aim of the work was to create enlarged variants of µdystrophins, called mididystrophins, in which, unstructured hinge regions, which have no therapeutic potential, were removed from a full-sized dystrophin.  

Three different mididystrophins, differing in the removed hinge regions, were created and analyzed in silico, in vitro and in vivo. Two of them demonstrated a good ability to form dystrophin-like molecular structures and had a functional and therapeutic effects when tested in cellular and animal models. It is also worth noting the good correlation between the results of AI-based structural modeling and the results of the experimental work,  which makes the work a good example of a successful balance of theoretical and experimental approaches, when the use of computational methods is very organically and competently integrated into the framework of extensive experimental work. In this regard, the work is worthy of being published in IJMS after correcting some shortcomings, to which I would attribute the following points.

1. The figure captions are very cumbersome and heavy, which is due to the fact that the authors repeat the same text three times for each mididystrophin variant. It would be desirable to change the structure of the captions, saying what is shown for 1(A), 2(B), and 3(C)…

2. The same applies to figures 1 and 3, in which the same color coding of pLDDT values ​​is given several times.

3. Some sections of the methods are written incomprehensibly, for example, 4.3. «GFP-GP41-1 split cassette design and protein-trans splicing assessment». There is a feeling that part of the text is missing, namely, it does not say how the genetic constructs used were obtained.

Comments on the Quality of English Language

minor editing

Author Response

The article is an original and interesting work demonstrating how using intein technologies and structural modeling with AlphaFoldF3 it is possible to create new therapeutically competent protein hybrids. The object of the study was the protein dystrophin, the absence of which leads to severe muscular dystrophies, which can be partially compensated by viral delivery of functionally important domains of dystrophin, called µdystrophins. The aim of the work was to create enlarged variants of µdystrophins, called mididystrophins, in which, unstructured hinge regions, which have no therapeutic potential, were removed from a full-sized dystrophin.  

Three different mididystrophins, differing in the removed hinge regions, were created and analyzed in silico, in vitro and in vivo. Two of them demonstrated a good ability to form dystrophin-like molecular structures and had a functional and therapeutic effects when tested in cellular and animal models. It is also worth noting the good correlation between the results of AI-based structural modeling and the results of the experimental work,  which makes the work a good example of a successful balance of theoretical and experimental approaches, when the use of computational methods is very organically and competently integrated into the framework of extensive experimental work. In this regard, the work is worthy of being published in IJMS after correcting some shortcomings, to which I would attribute the following points.

Comment 1: The figure captions are very cumbersome and heavy, which is due to the fact that the : authors repeat the same text three times for each mididystrophin variant. It would be desirable to change the structure of the captions, saying what is shown for 1(A), 2(B), and 3(C).

Response 1: We agree and we modified the figure legends as suggested.

Comment 2: The same applies to figures 1 and 3, in which the same color coding of pLDDT values ​​is given several times.

Response 2: We modified it as well

Comment 3: Some sections of the methods are written incomprehensibly, for example, 4.3. «GFP-GP41-1 split cassette design and protein-trans splicing assessment». There is a feeling that part of the text is missing, namely, it does not say how the genetic constructs used were obtained.

Response 3: We took into account the difficulty in reading such complicated names and we changed with a more generic title “ in vitro assessment of protein trans-plicing” where we described how we generated the reporter expression cassette and the studies in cell models (line 521).

Reviewer 3 Report

Comments and Suggestions for Authors

The manuscript of L. Palmieri et al. describes the design and generation of three dystrophin constructs, - Midi-Dys1, 2 and 3, to enable the use of similar approach in the treatment of Duchene Muscular Dystrophy (DMD). The dystrophin Midi-Dys 1 and 2 constructs included a central cytoskeleton-binding domain, nNos- and Part1-interacting domains, as well as a complete c-terminal region; Midi-Dys 3 contained no actin-binding site, while it was partially included in Midi-Dys 1 and 2.

The method used by the authors to obtain Midi-Dys constructs is based on double AAV transduction, allowing the final product to be obtained by cleaved splicing of inteine-based proteins, as which the authors chose GP41-1. Due to the limited capacity of AAV vectors, the authors reduced the molecular mass of the constructs introduced into the plasmid while making sure that the structural stability of the protein residues was maintained.

Using the deep-learning algorithm AlphaFold3, the authors predicted the possibility of interaction between two halves of split Midi Dystrophins in conjugation with intein. A strong interaction between the N- and C-termini of Midi-Dys1 and 2 was experimentally confirmed, while the similar interaction in Midi-Dys 3 was weak.

In experiments in the DBA2/mdx mouse model of DMD, the therapeutic effect of Midi-Dys1 and 2 constructs was confirmed, with no effect of Midi-Dys3.

In general, the work gives the impression of a well-designed, theoretically developed (in silico modelling) and carefully conducted experimental study. The methods of fluorescent microscopy (as well as its quantitation), flow cytometry, capillary Western Blot, as well as evaluation of dystrophy markers in animal models leave a very favourable impression.

The Discussion is written concisely and specifically. All experimental approaches used by the authors are adequately outlined in Materials and Methods, figure legends and Supplement.

Reviewer has no significant remarks to this manuscript.

Minor remark: The diagrams on the right side of Figure 1 D, E, F and Figure S1 B, C are not readable. Is it possible to improve their quality?

Author Response

The manuscript of L. Palmieri et al. describes the design and generation of three dystrophin constructs, - Midi-Dys1, 2 and 3, to enable the use of similar approach in the treatment of Duchene Muscular Dystrophy (DMD). The dystrophin Midi-Dys 1 and 2 constructs included a central cytoskeleton-binding domain, nNos- and Part1-interacting domains, as well as a complete c-terminal region; Midi-Dys 3 contained no actin-binding site, while it was partially included in Midi-Dys 1 and 2.

The method used by the authors to obtain Midi-Dys constructs is based on double AAV transduction, allowing the final product to be obtained by cleaved splicing of inteine-based proteins, as which the authors chose GP41-1. Due to the limited capacity of AAV vectors, the authors reduced the molecular mass of the constructs introduced into the plasmid while making sure that the structural stability of the protein residues was maintained.

Using the deep-learning algorithm AlphaFold3, the authors predicted the possibility of interaction between two halves of split Midi Dystrophins in conjugation with intein. A strong interaction between the N- and C-termini of Midi-Dys1 and 2 was experimentally confirmed, while the similar interaction in Midi-Dys 3 was weak. In experiments in the DBA2/mdx mouse model of DMD, the therapeutic effect of Midi-Dys1 and 2 constructs was confirmed, with no effect of Midi-Dys3.

In general, the work gives the impression of a well-designed, theoretically developed (in silico modelling) and carefully conducted experimental study. The methods of fluorescent microscopy (as well as its quantitation), flow cytometry, capillary Western Blot, as well as evaluation of dystrophy markers in animal models leave a very favourable impression.

The Discussion is written concisely and specifically. All experimental approaches used by the authors are adequately outlined in Materials and Methods, figure legends and Supplement.

Reviewer has no significant remarks to this manuscript.

Comment 1: The diagrams on the right side of Figure 1 D, E, F and Figure S1 B, C are not readable. Is it possible to improve their quality?

Response 1: We are grateful for the nice comments and of course, we corrected the little remarks.

Reviewer 4 Report

Comments and Suggestions for Authors

This is a well written study on the potential use of midi-dystrophin as a treatment for DMD. It combines the use of structural bioinformatics with experimental methods and provides a compelling case for the use of midi-dystrophins as a potential therapeutic for DMD.

Minor comments:

1) Section 4.2: Please make the custom Python code available on Github or some other similar website for researchers to use.

2) 4.12: Please make any R code available that was done to generate any graphs or analyses.

3) Can the authors comment in the Discussion on how far away approximately in terms of years midi-Dys is from actually being used therapeutically in humans? As in clinical trials of this approach? Or will this potential approach just disappear with this paper as only a proof of concept?

4) Are AAVs the best vectors for this? Or are there other vectors that are more efficient? Such as lentiviral or adenovirus vectors? Can the authors comment on this in the Discussion?

Author Response

This is a well written study on the potential use of midi-dystrophin as a treatment for DMD. It combines the use of structural bioinformatics with experimental methods and provides a compelling case for the use of midi-dystrophins as a potential therapeutic for DMD.

Minor comments:

Comment 1: Section 4.2: Please make the custom Python code available on Github or some other similar website for researchers to use.

Response 1: We thank the reviewer for the suggestion. The original code, as well as the amino-acid sequences are available here: GitHub - GNT-DDC/midi-dystrophin-af3: In silico structural prediction for the generation of novel performant Midi-dystrophins based on intein-mediated dual AAV approach (line 516).

Comment 2: Please make any R code available that was done to generate any graphs or analyses.

Response 2: in this study we did not use any R code for graphs and analysis. We used GraphPad Prism for making the graphs and statistic analysis.

Comment 3: Can the authors comment in the Discussion on how far away approximately in terms of years midi-Dys is from actually being used therapeutically in humans? As in clinical trials of this approach? Or will this potential approach just disappear with this paper as only a proof of concept?

Response 3: The use of dual-AAV intein-based gene therapy to deliver larger genes has rapidly progressed toward clinical application. This technology has already been employed in treating various genetic diseases in preclinical animal models. Recently, its in vivo application in humans, specifically for Usher syndrome type 1b (developed by Dr. Alberto Auricchio), has brought significant hope for the future use of this approach. Furthermore, with the latest results from clinical trials on micro-dystrophin showing only limited functional improvement in patients, the potential to create longer, more physiologically relevant and effective therapeutic genes, creates the basis to implement this technology in the clinics. However, the authors acknowledge that this research is still in its early stages and needs further optimization to ensure efficacy and safety in patients. Our group is actively working on this aspect.

The primary focus of this article is the discovery that in silico modeling using AlphaFold3 is a powerful tool for predicting protein trans-splicing (PTS) efficiency and it allowed the generation of two novel performant midi-dystrophin. Next preclinical studies will be directed towards the understanding of the advantage of our approach over the current ones, and optimization of safety and efficacy parameters.

We included these comments in the discussion (line 478).

Comment 4: Are AAVs the best vectors for this? Or are there other vectors that are more efficient? Such as lentiviral or adenovirus vectors? Can the authors comment on this in the Discussion?

Response 3: AAVs (adeno-associated viruses) are the most commonly used vectors for gene therapy due to several key advantages. They are non-integrative, meaning they don't cause integration-related toxicities. They can also be easily engineered for specific muscle targeting, as demonstrated in our recent work published in Nature Communications. Additionally, AAVs trigger much weaker immune responses compared to adenoviruses and lentiviruses, which is particularly important when high therapeutic doses are required, such as in DMD (Duchenne Muscular Dystrophy). In DMD treatment, achieving at least 40% dystrophin-positive myofibers (a major endpoint in clinical trials) necessitates high doses, and the use of an immune-toxic viral vector could amplify the risk of immune reactions against the therapeutic transgene.

We slightly described this concept better in the discussion (line 421).